# Anti-Cancer Effects of Oxygen-Atom-Modified Derivatives of Wasabi Components on Human Leukemia Cells

**DOI:** 10.3390/ijms24076823

**Published:** 2023-04-06

**Authors:** Jui-Feng Lin, Chih-Wen Chi, Yu-Chuen Huang, Tung-Hu Tsai, Yu-Jen Chen

**Affiliations:** 1Division of Neurosurgery, Department of Surgery, MacKay Memorial Hospital, Taipei 104, Taiwan; 2Institute of Traditional Medicine, School of Medicine, National Yang Ming Chiao Tung University, Taipei 112, Taiwan; 3Department of Medical Research, MacKay Memorial Hospital, Taipei 251, Taiwan; 4Department of Medical Research, China Medical University Hospital, Taichung 404, Taiwan; 5School of Chinese Medicine, College of Chinese Medicine, China Medical University, Taichung 404, Taiwan; 6Department of Radiation Oncology, MacKay Memorial Hospital, Taipei 251, Taiwan; 7MacKay Junior College of Medicine, Nursing, and Management, Taipei 112, Taiwan

**Keywords:** wasabi, 6-(methylsulfinyl)hexyl isothiocyanate, 6-(methylsulfenyl)hexyl isothiocyanate, 6-(methylsulfonyl)hexyl isothiocyanate, mitosis, autophagy, chronic myelogenous leukemia

## Abstract

1-Isothiocyanato-6-(methylsulfinyl)-hexanate (6-MITC) is a natural compound found in *Wasabia japonica*. The synthetic derivatives 1-Isothiocyanato-6-(methylsulfenyl)-hexane (I7447) and 1-Isothiocyanato-6-(methylsulfonyl)-hexane (I7557) were obtained from 6-MITC by deleting and adding an oxygen atom to the sulfone group, respectively. We previously demonstrated that extensive mitotic arrest, spindle multipolarity, and cytoplasmic vacuole accumulation were induced by 6-MITC and inhibited the viability of human chronic myelogenous leukemia K562 cells. In this study, we examined the anti-cancer effects of 6-MITC derivatives on human chronic myelogenous leukemia (CML) cells. Autophagy was identified as the formation of autophagosomes with double-layered membranes using transmission electron microscopy. Cell cycle and differentiation were analyzed using flow cytometry. Apoptosis was detected by annexin V staining. After treatment with I7447 and I7557, the G2/M phase of cell cycle arrest was revealed. Cell death can be induced by a distinct mechanism (the simultaneous occurrence of autophagy and aberrant mitosis). The expression levels of acridine orange were significantly affected by lysosomal inhibitors. The natural wasabi component, 6-MITC, and its synthetic derivatives have similar effects on human chronic myelogenous leukemia cells and may be developed as novel therapeutic agents against leukemia.

## 1. Introduction

Chronic myeloid leukemia (CML) is an acquired disorder with no known inherited predisposition. However, disease concordance is absent in monozygotic twins. The Philadelphia chromosome, a shortened version of chromosome 22, characterizes the disorder. The Philadelphia chromosome is a translocation between chromosomes 9 and 22 that breaks the long arms at q34 and q11 [1,2,3]. Myeloid leukemia is usually detected in the chronic phase, and the clonal expansion of mature myeloid cells characterizes CML. All untreated patients eventually progress to a lethal blast phase, which is occasionally preceded by an accelerated phase. Chemotherapy, interferon therapy, stem cell transplantation, differentiation induction, and targeted therapy can treat the different phases of CML [4,5]. The response rates for targeting therapy, interferon therapy, and stem cell transplantation are 80–90%, 33–58% and 59–94% and the 5-year overall survival rates are 90%, 34–90% and 60%, respectively [5,6,7,8]. The resistance rate to TKIs ranged from 40 to 80% [5]. The median survival time following chemotherapy ranged from 33 to 55 months [9]. The response rate and survival rate might need further research for differentiation induction therapy [10]. The BCR (breakpoint cluster region) and ABL (Abelson murine leukemia viral oncogene homolog 1) genes play important roles in CML [11]. The BCR-ABL fusion gene encodes a chimeric protein that enhances the protein tyrosine kinase and autophosphorylation activity of ABL as well as the relocalization of the kinase to cytoplasm, resulting in a constitutively active ABL kinase. It transforms hematopoietic stem cells into leukemic stem cells and activates the overproduction of leukocytes in the bone marrow [12]. The chimeric protein is expressed from the derivative 9q + chromosome in 70% of patients with CML [11]. As a result, BCR-ABL has been recognized as the most important target for CML treatment [13]. Therapies targeting tyrosine kinases that block BCR-ABL transcript expression are the most promising strategies for treating patients with CML who are ineligible for bone marrow transplantation as a treatment. Imatinib mesylate (IM; also known as STI-571) is a first-line drug developed for CML that targets tyrosine kinases. The estimated overall survival rate at 10 years for patients using this drug is 83.3% and the complete cytogenetic response is 82.8% [14]. Resistance to IM in patients with CML harboring BCR-ABL mutations is associated with several clinical issues [15,16]. The resistance rates to imatinib in accelerated-phase CML are 45% and 75% after 2 and 4 years of imatinib therapy, respectively [17]. The 4-year resistance rates of the later chronic and accelerated blastic phases are 20% and 70–90%, respectively [18,19]. Therefore, novel drugs other than BCR-ABL inhibitors are urgently needed to treat CML. Currently, dasatinib may produce a better outcome than imatinib in the first 12 months [16], and asciminib could be used as a third-line therapy to treat CML [20,21]. Since imatinib was the first TKI targeting BCR-ABL in the treatment of CML, the imatinib-resistant CML K562 cell line was established in our laboratory for experiments. The long-term effects of tyrosine kinase inhibitors are cardiotoxicity [22], and bone pain [23].

The compound 1-Isothiocyanato-6-(methylsulfinyl)-hexanate (6-MITC), derived from *Wasabia japonica*, has anti-metastasis [24], chemopreventive [25], and anti-cancer [25] effects, prevents bone loss [26] and provides protection of hepatocytes [27]. These published studies suggest that 6-MITC and its derivatives may have the potential to exhibit less toxicity to the heart and bone in the treatment of CML. We previously demonstrated the viability of human CML K562 cells and induced extensive mitotic arrest, spindle multipolarity, and cytoplasmic vacuole accumulation in the presence of 6-MITC [28]. Notably, cell death (through the concurrent induction of mitosis and autophagy) was induced by 6-MITC. These results revealed that 6-MITC has therapeutic potential. The synthetic derivatives 1-Isothiocyanato-6-(methylsulfenyl)-hexane (I7447: sulfide containing no oxygen) and 1-Isothiocyanato-6-(methylsulfonyl)-hexane (I7557: sulfone containing two oxygens) were prepared from 6-MITC (sulfone containing one oxygen) by the deletion and addition of oxygen, respectively, and showed different effects on human oral cancer cells (Figure 1) [29]. Although the structure–activity relationships of these three compounds may reveal varying degrees of antitumor activity, the therapeutic roles of I7447 and I7557 in leukemia remain unclear. Furthermore, the structure–activity relationship of these three compounds in leukemia cells remains to be determined. In this study, we sought to determine the effects of the 6-MITC derivatives I7447 and I7557 on the induction of aberrant mitosis and autophagy in leukemia cells.

## 2. Results

### 2.1. Growth Inhibition of I7447-, 6-MITC-, and I7557-Treated Human K562 and HEL Cells

The K562 and HEL cells were treated with 2.5, 5, or 10 μM I7447, 6-MITC, or I7557 for 48 h. These compounds inhibited the viability of K562 and HEL cells in a dose-dependent manner (Figure 2A,B). The half-maximal inhibitory concentration (IC_50_) values of I7447, 6-MITC, and I7557 at 48 h were 4.00, 4.12, and 4.96 μM, respectively, in K562 cells and 5.50, 1.43, and 1.06 μM, respectively, in HEL cells. For the viability of IM-resistant K562 cells, IM had no inhibitory effect. The wasabi component 6-MITC and its derivative, I7557, suppressed the viability of IM-resistant K562 cells, indicating their effectiveness on both parent and IM-resistant K562 cells (Figure 3).

### 2.2. Impact of I7447, 6-MITC, and I7557 on the Cell Cycle

To determine the effects of I7447, 6-MITC, and I7557 on human leukemia cells, we evaluated the cell cycle progression in human K562 and HEL cells. Based on the cell cycle analysis of K562 cells, 6-MITC and I7557 induced significant G2/M phase arrest in a concentration-dependent manner, whereas I7447 and I7557 caused significant sub-G1 phase arrest (Figure 4A,B). In addition, I7447, 6-MITC, and I7557 reduced the proportion of HEL cells in the G0/G1 phase concentration-dependently (Figure 4C,D). For K562 cells treated with I7557, further classification of the G2/M arrest by using expression of phosphorylated Histone H3 showed its increased expression, indicating arrest at the mitosis phase (Appendix A). I7557 treatment for 48 h down-regulated the expression of p-Chk1, p-Chk2, p-cdc25c, and p-cdc-2 (Appendix A), whereas the expression of p-plk1 was upregulated (Appendix A). These results implicated proteins related to mitosis phase arrest in the mechanistic action of I7557.

### 2.3. Exclusion of Cell Differentiation by I7447, 6-MITC, and I7557

To clarify whether I7447, 6-MITC, and I7557 could trigger the differentiation of K562 and HEL cells, the specific surface antigens CD14, CD16, CD61, and CD235a, which are associated with the differentiation of monocytes, granulocytes, megakaryocytes, and erythrocytes, respectively, were monitored by flow cytometry. Compared with those in the control cells, I7447, 6-MITC, and I7557 did not alter the expression levels of these surface antigens in treated cells (Figure 5A,B).

### 2.4. Autophagy in I7447-, 6-MITC-, and I7557-Treated Human K562 Cells

Several methods have been employed to investigate modes of cell death, including autophagy and apoptosis. After treating K562 and HEL cells with I7447, 6-MITC, and I7557, extensive mitotic arrest, spindle multipolarity, and cytoplasmic vacuoles were observed by Liu staining and light microscopy (Figure 6A,B). Autophagosomes with double-layered membrane structures containing organelle remnants were observed in the accumulated cytoplasmic vacuoles by transmission electron microscopy (TEM) (Figure 6C). Using flow cytometry, acridine orange staining of acidic vesicular organelles indicated the acidity of the autophagosomes (Figure 6D). Expression levels of acridine orange were elevated by treatment with I7447, 6-MITC, and I7557. The lysozyme inhibitor E/P (10 μg/mL) further increased the expression level of acridine orange in treated cells, implying an influence on autophagic flux (Figure 6E). The expression profiles of the autophagic factors m-TOR, beclin-1, ATG 5, ATG 7, ATG 12, Bcl-2, and BNIP 3were evaluated by immunoblotting after treatment with I7447, 6-MITC, and I7557 (Figure 6F). No significant changes were observed in the expression levels of these proteins.

### 2.5. Cell Viability and Cell Cycle Analysis after Treatment with an Autophagy Inhibitor

To determine whether I7447, 6-MITC, and I7557 induced the autophagy process necessary for cell cycle blockage, the autophagy inhibitors 3-methyladenine (3-MA; 2 mM), chloroquine (CQ; 12.5 μM), E64D/pepstatin A (E/P; 10 μM), bafilomycin A1 (BafA1; 20 nM), and rapamycin (40 nM) were used. The cell cycle and cell viability were analyzed. K562 cell growth was inhibited by treatment with I7447, 6-MITC, or I7557. These autophagy inhibitors influenced neither cell viability nor the cell cycle distribution (Figure 7).

### 2.6. Apoptosis Induction by I7447 and I7557

Before flow cytometric analysis, apoptosis was detected by staining cells with annexin V and propidium iodide to assess apoptotic and necrotic cells [30]. Treatment with I7447 and I7557 for 48 h induced apoptosis in a concentration-dependent manner, as shown in Figure 8A,B.

## 3. Discussion

The findings of this study indicate that I7447 and I7557 have similar effects on 6-MITC in human leukemia K562 and HEL cells. Aberrant mitosis, autophagy, and apoptosis were observed in the compound-treated cells. Furthermore, autophagic cell death was observed under the influence of autophagic flux. Autophagy is an evolutionarily conserved intracellular process in which cytoplasmic components, including entire organelles, are targeted for lysosomal degradation [31]. Autophagic vacuoles accumulate toxic peptides and serve as intracellular reservoirs. However, their removal is inefficient because of impaired maturation. Autophagy malfunction is associated with various diseases, including cancer, metabolic, and neurodegenerative diseases [32]. In colon cancer, wasabi induces autophagy by decreasing the phosphorylation of Akt and mTOR and promotes the expression of microtubule-associated protein 1 light chain 3-II and AVO formation [33]. Lysosomes are regarded as mediators of energy metabolism and cellular clearance as they control autophagy and the metabolic signaling molecules AMPK and mTORC1 [34,35]. Autophagy initiation is indicated by developing a double-layered crescent-shaped membrane known as a phagophore, which elongates and matures into an autophagosome. Autophagosomes engulf and sequester long-lived proteins and damaged organelles, which are then degraded in the lysosomes. This process is referred to as autophagic flux. Blocked autophagic flux is characterized by the accumulation of undigested macromolecules and autolysosomes, which can result in enlarged lysosomes and aggravation of lysosomal injury [36]. In L1210 cells, cell death effects can be induced by sulforaphane and allyl isothiocyanate [37]. Apoptosis and necrosis were detected using annexin V and propidium iodide staining, and autophagy was detected using monodansylcadaverine [37]. Apoptosis and autophagy are involved in cell death. Therefore, autophagy inhibitors were used to identify the possible mechanisms underlying induced autophagy.

Cell cycle arrest at the G2/M phase was assessed using flow cytometry. Our previous study revealed high levels of histone H3 phosphorylation, which indicated cell cycle arrest in the M phase by 6-MITC [28]. Therefore, we postulated that the chemical derivatives of 6-MITC might have similar effects on the M phase. Through the inhibition of the nuclear factor-kappa B (NF-κB) pathway, the disturbance of mitochondrial function and the induction of apoptosis in different cancer cells have been observed. The antitumor characteristics of 6-MITC have been reported [38,39,40,41]. Flow cytometric analysis revealed a higher percentage of cells in the sub-G1 phase (apoptotic cells), particularly in I7447-, 6-MITC-, and I7557-treated K562 cells, indicating increased levels of autophagy or apoptosis. In human pancreatic cancer cells, 6-MITC and I7557 induce G2/M phase arrest and a hypoploid population [40]. In oral cancer cells, cell cycle analysis revealed a considerable G2/M arrest in 6-MITC- and I7557-treated cells, whereas sub-G1 accumulation was observed in I7447-treated cells [29]. Imatinib binds to the amino acids of the BCR/ABL tyrosine kinase ATP-binding site to prevent tyrosine autophosphorylation, which contributes to influence tumor growth [14]. The 6-MITC compound and its derivatives could induce G2/M phase arrest in various kinds of tumor cells and then promote cell death by both autophagic cell death and apoptosis. The difference in signaling targets might be related to the cell cycle control machinery or DNA damage repair mechanisms. Further elucidation of these mechanisms is needed.

In this study, I7557, 6-MITC, and I7447 showed potent autophagy-inducing effects in chronic leukemia cells. This is the first study to publish research in this field. In this study, the effects of I7447- and I7557-induced autophagy were not reversed by the autophagy inhibitors 3-MA, CQ, E/P, BAFA1, and rapamycin. These findings indicate that I7447 and I7557 induce autophagic cell death via distinct mechanisms. Therefore, other mechanisms must be elucidated. Since the lysozyme inhibitor E/P increased the expression level of acridine orange in treated cells, this implies that signaling related to the autophagic flux or fusion of lysosome and autophagosome might be potential mechanisms. Annexin V staining revealed apoptosis as cell death induced by I7447 and I7557. The expression levels of acridine orange in the acidic vacuoles were significantly affected, indicating that I7447, 6-MITC, and I7557 blocked autophagic flux and induced autophagy in K562 cells. The differential pharmacological effects of 6-MITC and its chemical derivatives, which consist of different numbers of oxygen atoms, may be similar to those in human oral cancer cells [29]. In treated cells compared with control cells, the expression of the specific surface antigens CD14, CD16, CD61, CD41a, CD42b, and CD235a is associated with cell differentiation into monocytes, granulocytes, megakaryocytes, platelets, and erythrocytes. The flow cytometry results showed that I7447, 6-MITC, and I7557 did not alter the differentiation of K562 and HEL cells. These findings indicate that chemical derivatives of 6-MITC induce autophagy, apoptosis, and G2/M phase arrest but not differentiation. According to the cell cycle data and acridine orange stain results, I7557 is more active than 6-MITC and I7447. There is no difference in the activity of apoptosis induction between I7447 and I7557. The possible cause of differential efficacy by these three compounds might be the different oxygen atom numbers on the sulfide group. Other than altering oxygen numbers on 6-MITC, reports about activity alteration by the addition or removal of oxygen atoms is lacking in the literature. The most similar concept was noted in the replacement of oxygen atoms by other atoms such as sulfur atoms [42]. In several bacteria, when a nonbridging oxygen was replaced by a sulfur atom in synthetic phosphorothioate internucleotide linkages, the modified structures shared similar physical and chemical properties with phosphodiesters but exhibited enhanced DNA/RNA tolerance towards nucleases [42]. It may be desirable to start focusing on the design and evaluation of a proper structure–activity relationship (SAR) to improve potency and to elucidate a possible mechanism of action.

The limitations of this study include the lack of mechanistic insight into the apoptosis induced by these compounds. However, this is a minor mode of death. Another limitation is that data from only three compounds may be insufficient to provide comprehensive information for interpreting the structure–activity analysis. The novelty of this study is that the demonstration that the number of oxygen atoms may play a role in the bioactivity of 6-MITC derivatives, and that 6-MITC could be used as the main structural backbone to modify and synthesize new derivatives for the further development of potential anti-cancer compounds.

## 4. Materials and Methods

The methods used here are similar to those used in our previous work [14]. In addition, new materials, I7447 and I7557, synthetic derivatives of 6-MITC, were prepared for this study.

### 4.1. Chemicals

Pure 6-MITC, I7447, and I7557 were purchased from LKT Laboratories (St. Paul, MN, USA). The compounds were prepared in dimethyl sulfoxide (DMSO) (Merck, Darmstadt, Germany) and stored at −20 °C. There is no significant change in cell viability about the comparison of control group with vehicle group (DMSO) in preliminary study (Appendix A). The DMSO was added in each control group alone, except E/P solution. The E/P was dissolved in ethanol. The control group of E/P was combined with DMSO and ethanol. The concentration of DMSO in the culture medium is 0.1%, but 0.1% DMSO and 0.1% ethanol in the culture medium for group E/P.

### 4.2. Cell Lines and Culture

Human erythroleukemia HEL and human CML K562 cells were obtained from the American Type Culture Collection (Manassas, VA, USA) and cultured in RPMI 1640 medium (Gibco, Grand Island, NY, USA) supplemented with 10% fetal bovine serum (Gibco) and 2 mM L-glutamine (Sigma-Aldrich, St. Louis, MO, USA). The cells were subcultured every 2 to 3 days to maintain the exponential growth phase. The IM-resistant K562 cells were established in our previous study and has a greater IC_50_ in comparison with K562 cells [43].

### 4.3. Cell Viability

The HEL (2 × 10^5^/mL) and K562 (1 × 10^5^/mL) cells were treated with different concentrations (2.5, 5, and 10 μM) of 6-MITC, I7447, and I7557. Briefly, 3-MA (1 or 2 mM) [44], CQ (12.5 μM) [45], E/P (10 μM) [46], BafA1 (20 nM) [47], or rapamycin (40 nM) [48] were added to the cells as autophagy initiators and cultured for 1 h, followed by 6-MITC (10 μM), I7447 (10 μM), and I7557 (10 μM) and cultivated for a further 48 h. The cells were harvested after treatment, and the number of viable cells was estimated using the trypan blue dye exclusion assay; the results were used to estimate cell viability [49].

### 4.4. Morphology Based on Light and Transmission Electron Microscopy

Light and electron microscopes were used to assess cell morphology, following the procedures modified from our previous study [28].

Cell morphology was observed under a light microscope (Olympus, Tokyo, Japan) at 1000× magnification. The cells were stained by Liu’s staining method; using Liu A solution for 45 s, followed by Liu B solution for 90 s. Cells were collected, washed, and fixed with 2.5% glutaraldehyde in cacodylate buffer for 30 min for TEM. Cells were fixed with osmium tetroxide (1%) and embedded in Epon resin (Electron Microscopy Science, Hatfield, PA, USA). Samples were stained with 0.5% toluidine blue and examined under a light microscope. Semi-thin and ultra-thin sections were cut. Ultra-thin sections were stained with 2% uranyl acetate and Reynold’s lead citrate and visualized using TEM with a digital camera (JEM-1200EXII, JEOL Co., Tokyo, Japan).

### 4.5. Cell Cycle Analysis Using a DNA Histogram

The DNA staining was conducted using a BD Cycletest^TM^ Plus DNA Reagent Kit (BD Biosciences, Franklin Lakes, NJ, USA) according to the manufacturer’s protocol. After treatment, the harvested cells were fixed with 70% ethanol at 4 °C for 1 h. The cells were then incubated with solutions A and B, each containing 0.1 mg/mL RNase, for 30 min and 0.5 mg/mL propidium iodide (PI) for 10 min. Following filtration using a 50-μm nylon mesh, the cells were analyzed using a FACSCalibur flow cytometer (Becton Dickinson, Lincoln Park, NJ, USA). In addition, the observed DNA histogram was evaluated. Data from 10^4^ cells were acquired and analyzed using the ModFit software 3.0 (Becton Dickinson).

### 4.6. Surface Antigen Assay for Differentiation

Indirect immunofluorescence was used to detect the expression of differentiation- associated antigens on the surface of leukemic cells. Cells on days 1, 3, and 5 were washed with phosphate buffered saline (PBS) and incubated with fluorescein isothiocyanate (FITC)-conjugated secondary antibodies (goat F(ab’)2 anti-mouse IgG, (Cappel Laboratories, Cochranville, PA, USA) with primary monoclonal antibodies. Differentiated antigens were used to detect differentiation. Examples included anti-CD14 (BD Biosciences) for monocytes [50], anti-CD16 [51] (Miltenyi Biotec, Bergisch Gladbach, Germany) for neutrophils, anti-CD235a (BD Biosciences) for erythrocytes, and anti-CD61 [52] (BD Biosciences) for megakaryocytes. The background threshold was obtained using a FITC-conjugated goat anti-mouse IgG antibody. Flow cytometry was conducted using a FACSCalibur with an Argon-ion laser and a 488 nm emission filter, and the data were analyzed using the CellQuest^Pro^ software 5.1 (Becton Dickinson).

### 4.7. Acridine Orange Stain

The cells were stained with acridine orange (10 ng/mL) for 15 min and subjected to flow cytometry to quantify the development of acidic vesicular organelles. The cytoplasm and nucleoli emitted green fluorescence in cells stained with acridine orange. In contrast, the acidic compartments emitted red fluorescence proportional to the degree of acidity. Red (650 nm) and green (510–530 nm) fluorescence signals were emitted by 10^4^ cells illuminated with blue excitation light (488 nm). The signals were recorded using a Samea flow cytometer.

### 4.8. Western Blot Analysis

After I7447 (10 μM), 6-MITC (10 μM), and I7557 (10 μM) were used to treat K562 cells, total protein was extracted and quantified using a bicinchoninic acid protein assay kit (Bio-Rad Laboratories, Hercules, CA, USA). Proteins were separated using 10% sodium dodecyl sulfate-polyacrylamide gel electrophoresis and transferred to a nitrocellulose membrane. Blotting was performed using various primary and horseradish peroxidase-conjugated secondary antibodies (1:5000; Santa Cruz Biotechnology, Dallas, TX, USA). ImageJ software estimated relative protein intensity using densitometry (version 1.36b, NIH, Bethesda, MD, USA). The mean values of the intensities were calculated from at least three independent experiments and normalized to that of β-actin.

### 4.9. Detection of Apoptosis with Annexin V Staining

The cells were stained using a TACS Annexin V-FITC Apoptosis Detection Kit (R&D Systems, Minneapolis, MN, USA) according to the manufacturer’s protocol. Briefly, cells were seeded at a density of 1 × 10^5^/mL and treated with I7447 (10 μM), 6-MITC (10 μM), or I7557 (10 μM) for 48 h. Cells (5 × 10^5^) were collected, washed with PBS, and resuspended in 100 μL of the annexin V incubation reagent. The mixtures were then incubated in the dark for 15 min at 25 °C. Finally, 400 μL of 1× binding buffer was added to each sample, which was immediately analyzed using a FACSCalibur flow cytometer.

### 4.10. Immunofluorescent Staining

The harvested cells were fixed in 4% paraformaldehyde for 10 min and permeabilized with 1% Triton X-100 in PBS. The cells were incubated in 10% bovine serum albumin (BSA) and anti-LC-3B (Abcam #ab48394, Cell Signaling Technology, Danvers, MA, USA) at a 1:400 dilution after serial washes with PBS. Cells were washed with PBS and incubated with the secondary antibody, rhodamine Red^TM^-X-conjugated goat anti-rabbit IgG (1:200; Jackson ImmunoResearch Laboratories, Inc., West Grove, PA, USA). The cells were then incubated with Hoechst 33342 (Sigma-Aldrich) to identify the cell nuclei.

### 4.11. Statistical Analysis

The results are given in terms of mean ± standard error of the mean (SEM). The results of each experiment were compared using a one-way analysis of variance as indicated. A statistical *p* value < 0.05 indicated statistical significance.

## 5. Conclusions

In conclusion, G2/M phase arrest could be induced by I7447 and I7557 in human CML cells. They can induce cell death in a distinct manner (autophagy and aberrant mitosis), similar to 6-MITC. The synthetic compounds I7447 and I7557 (synthesized by the deletion and addition of oxygen to sulfone) are derived from the naturally occurring *Wasabia japonica* component 6-MITC and can be used to treat CML. The structure–activity relationship with different oxygen atom numbers on the sulfide group to induce cell death in CML cells is unique and could be applied to develop new therapeutics against CML.

## Figures and Tables

**Figure 1 ijms-24-06823-f001:**
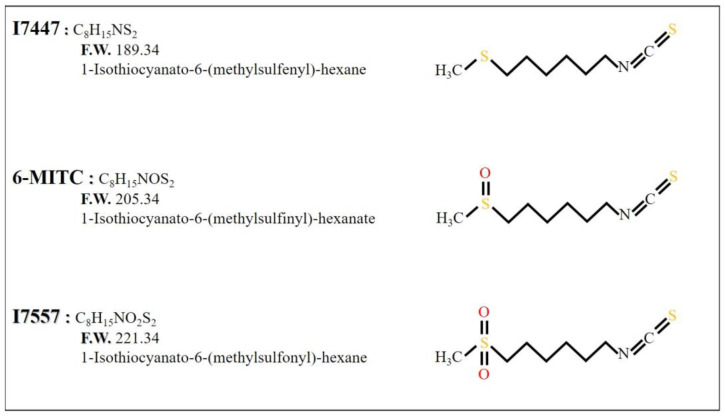
Chemical structures of 6-MITC, I7447, and I7557.

**Figure 2 ijms-24-06823-f002:**
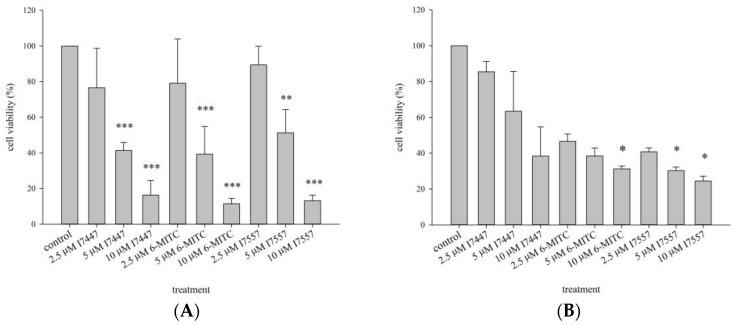
Effects of I7447, 6-MITC, and I7557 on the viability of K562 and HEL cells. Cell viability was assessed using the trypan blue exclusion assay. K562 and HEL cells were treated with or without 10 μM I7447, 6-MITC, and I7557 for 48 h. (**A**) K562; and (**B**) HEL cells. Data from three separate experiments are expressed as the mean ± standard error of the mean (SEM). N = 3 for each group. * *p* < 0.05, *** p* < 0.01, **** p* < 0.001.

**Figure 3 ijms-24-06823-f003:**
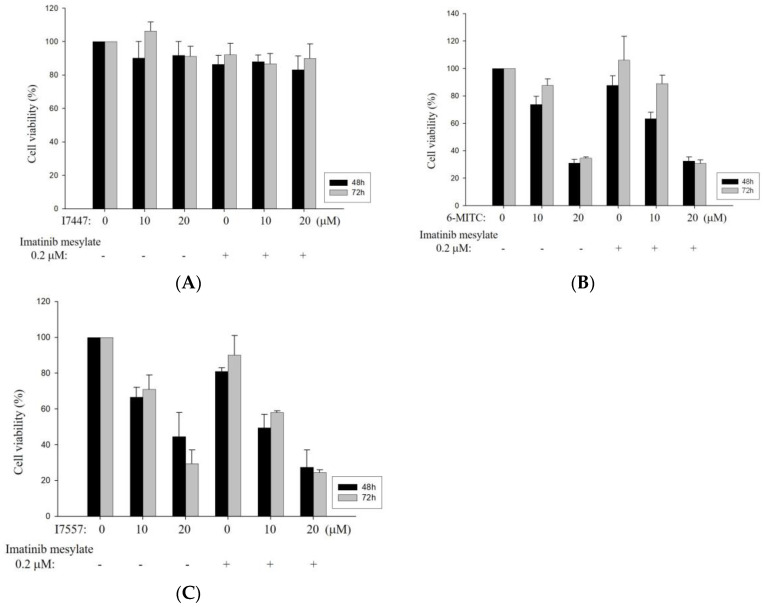
Effect of wasabi compounds on viability of parent and imatinib−resistant K562 cells. Cell viability was assessed using the trypan blue exclusion assay. K562 cells were treated with or without 0.2 μM imatinib mesylate for 48 and 72 h. (**A**) I7447; (**B**) 6-MITC; and (**C**) I7557 cells. Data from three separate experiments are expressed as the mean ± standard error of the mean (SEM). N = 3 for each group.

**Figure 4 ijms-24-06823-f004:**
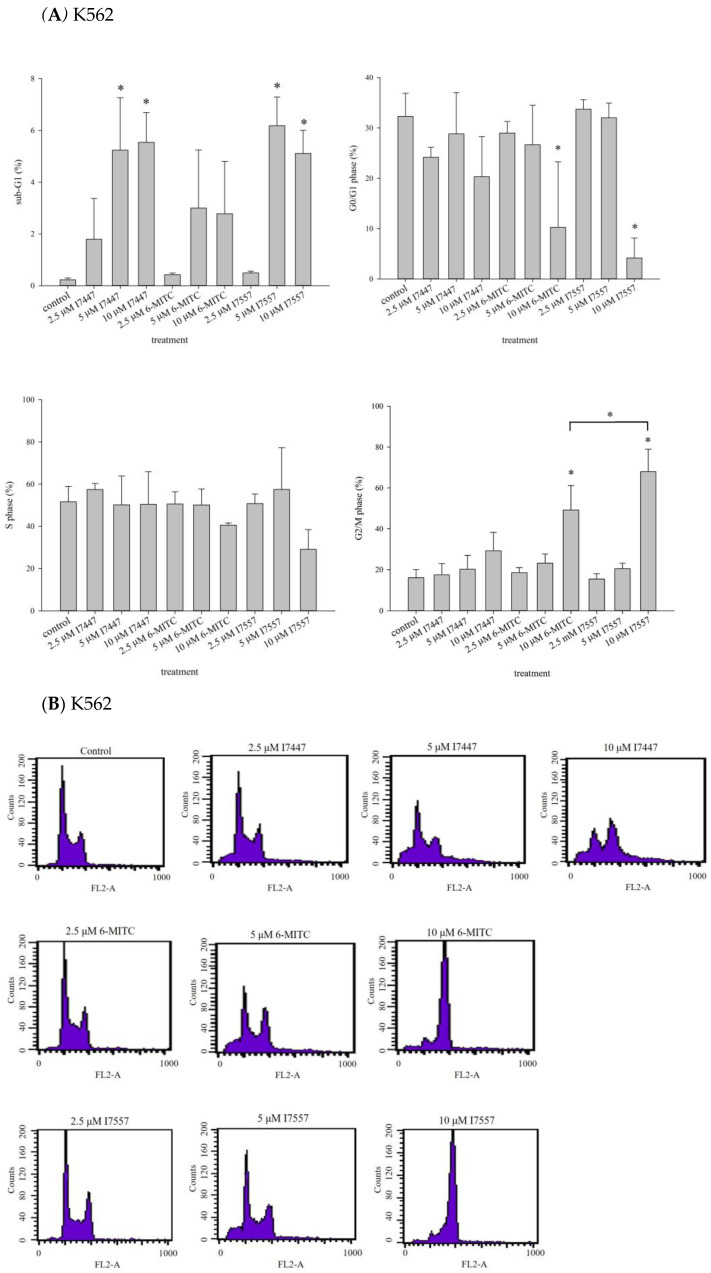
Cell cycle analysis of K562 and HEL cells treated with 6-MITC, I7447, and I7557. Expression level and representative DNA histogram at the sub-G1 and G2/M phases for K562 (**A**,**B**) and HEL (**C**,**D**) cells. Data from three separate experiments are expressed as the mean ± standard error of the mean (SEM). N = 3 for each group. * *p* < 0.05, *** p* < 0.01, **** p* < 0.001.

**Figure 5 ijms-24-06823-f005:**
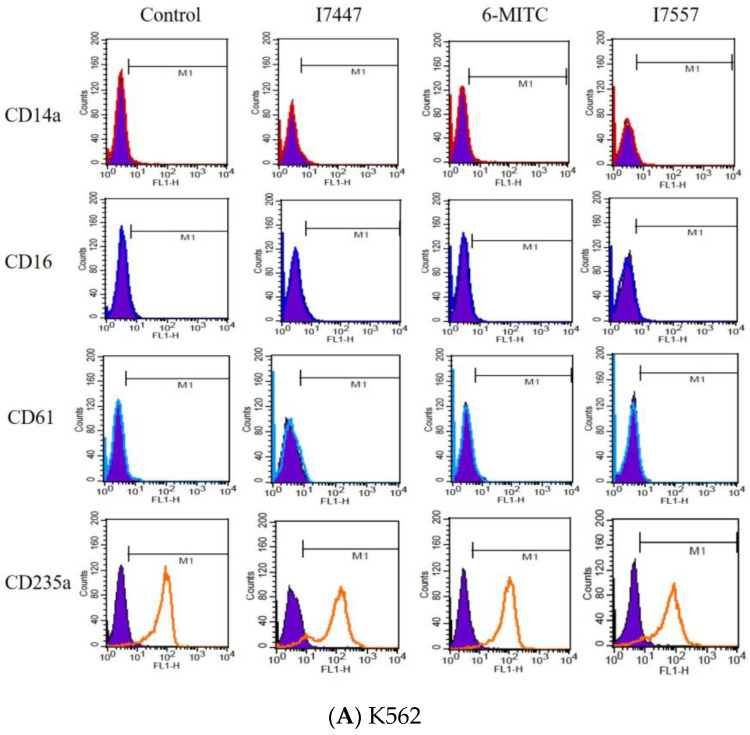
Expression of surface antigens on K562 and HEL cells after exposure to I7447, 6-MITC, and I7557. I7447, 6-MITC, and I7557 were incubated for 48 h and assessed by flow cytometry. (**A**) K562 cells; and (**B**) HEL cells. The expression levels of these surface antigens were analyzed using the CellQuest Pro software.

**Figure 6 ijms-24-06823-f006:**
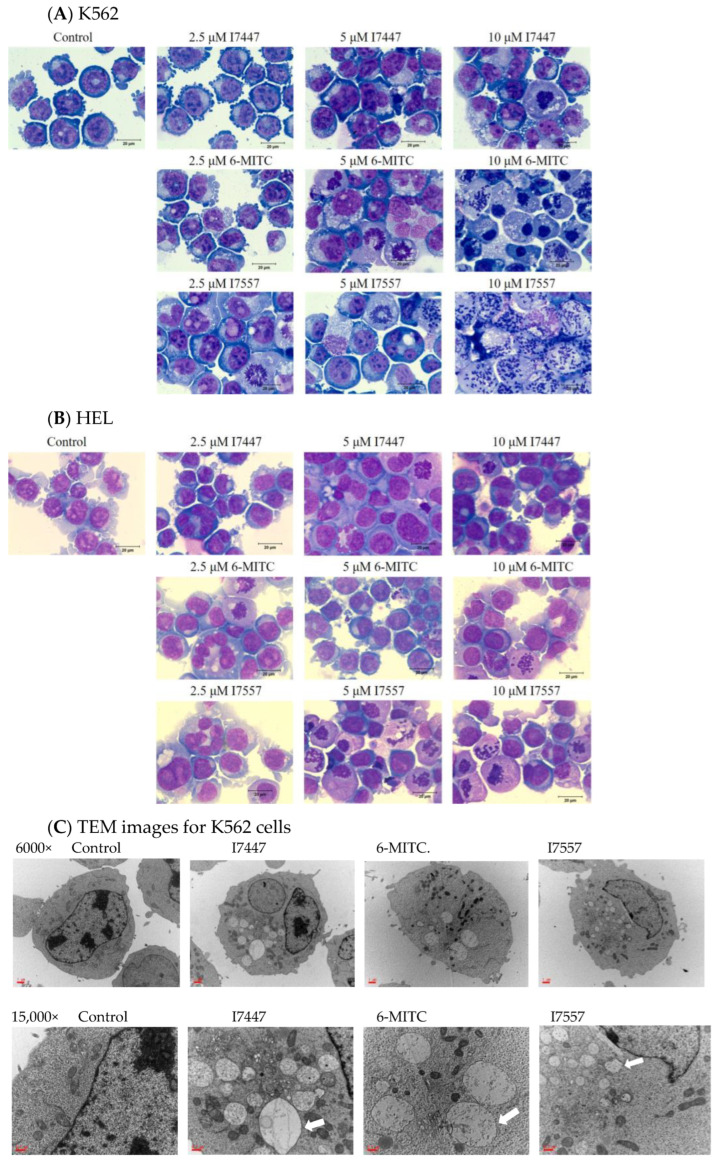
Autophagy in I7447-, 6-MITC-, and I7557-treated human K562 cells. After Liu’s staining, images were captured at 1000× magnification using a light microscope: (**A**) K562 cells; (**B**) HEL cells. (**C**) TEM images of K562 cells. The accumulated cytoplasmic vacuoles were identified as autophagosomes (indicated by arrows) with double-layered membrane structures containing remnants of organelles. (**D**) Using flow cytometry, acridine orange staining for acidic vesicular organelles was estimated. The expression levels of acridine orange were elevated by treatment with I7447, 6-MITC, and I7557. (**E**) the autophagy inhibitor E/P (10 μg/mL), further increased the expression levels of acridine orange in the treated cells. (**F**) The expression profile of the autophagy-related proteins was evaluated by immunoblotting after treatment with I7447, 6-MITC, and I7557. Data from three separate experiments are expressed as the mean ± standard error of the mean (SEM). N = 3 for each group. * *p* < 0.05, **** p*< 0.001.

**Figure 7 ijms-24-06823-f007:**
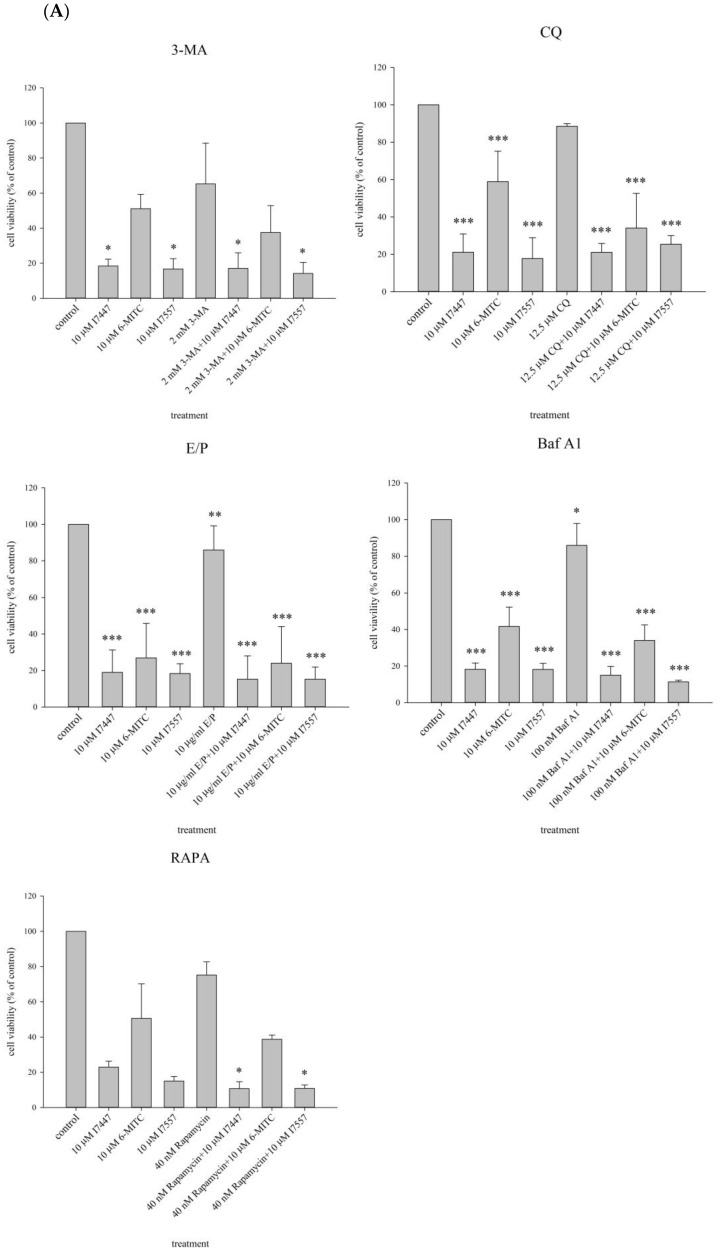
Cell viability and cell cycle analysis of K562 cells treated with 6-MITC, I7447, and I7557. 6-MITC, I7447, and I7557 were used to treat K562 cells. 3-methyladenine (3-MA; 2 mM), chloroquine (CQ; 12.5 μM), E64D/pepstatin A (E/P) (10 μM), bafilomycin A1 (BafA1; 20 nM), and rapamycin (40 nM) were used. Cell viability (**A**) and cell cycle (**B**) were assessed using the trypan blue exclusion assay and flow cytometry, respectively. Data from three separate experiments are expressed as the mean ± standard error of the mean (SEM). N = 3 for each group. * *p* < 0.05, *** p* < 0.01, **** p* < 0.001.

**Figure 8 ijms-24-06823-f008:**
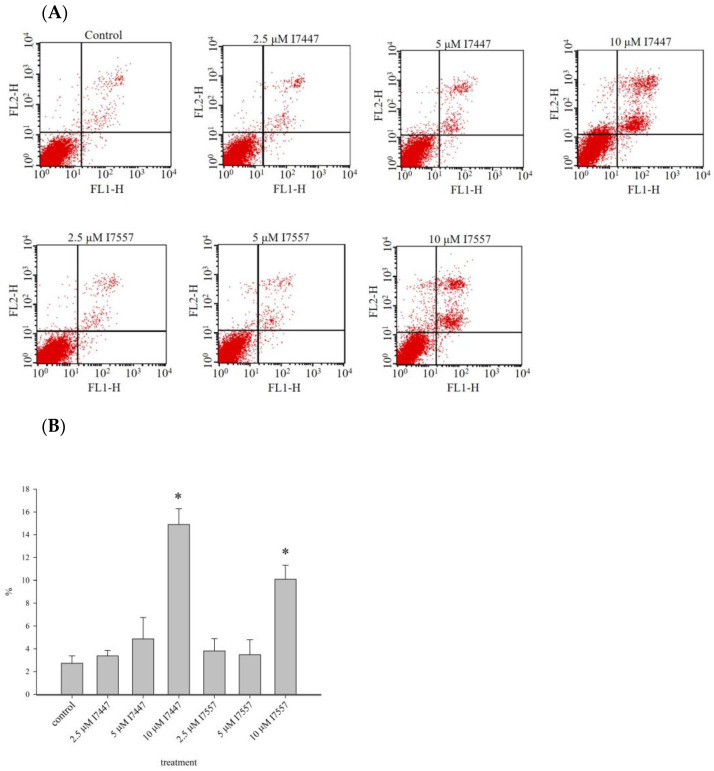
Apoptosis induction by I7447 and I7557. Based on flow cytometry analysis following annexin V staining, I7447 (10 μM) and I7557 (10 μM) induced significant apoptosis. (**A**) Representative flow cytometric figures. (**B**) Quantitative data for apoptotic cells with positive annexin V staining. Data from three separate experiments are expressed as the mean ± standard error of the mean (SEM). N = 3 for each group. * *p* < 0.05.

## Data Availability

The data and materials used to support the findings of this study are included in this article.

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
