# Peer review of "Anti-Cancer Effects of Oxygen-Atom-Modified Derivatives of Wasabi Components on Human Leukemia Cells"

_ijms, 2023, doi:10.3390/ijms24076823_

Round 1

Reviewer 1 Report (New Reviewer)

The compounds were prepared in dimethyl sulfoxide (DMSO) solution, but I couldn't find any control group with maximum used DMSO concentration. Could you please comment did you do control experiments with DMSO and how can you prove that the effects on cells you shown are due to derivatives of wasabi components and not the solution itself? 

Author Response

Reviewer 2 Report (New Reviewer)

Introduction:

-Line 51: add effectiveness of each of the therapies on CML patients recovery, resitance rate  and 5-years receding.

-Define BCR (breakpoint cluster region gene) and ABL (Abelson murine leukemia viral oncogene homolog 1) also Tyrosine kinase generalities in this context.

-Authors focus on citing Imitanib as TK inhibitor, please add same info for other inhibitors such as Dasatinib and Asciminib and compare. Also, could be useful for general scientific audience to know why you focus on Imitanib.

-Figure 1: please show sulfur atom and oxygen atoms in MITC and its two derivativesin another color.

-Add info about Imitanib long-term effects (post-recovery patients and children). Are MITC and derivatives expected to have less effects on other kinases, liver toxicity, skeletal growth, and why?

-Add litterature about modifiying active compounds with oxygen addition/removal and effects,

Results:

-Figure 3: any data after 72h? does it show a trend of restoration from cell death?

-Figure 6 A and B: add scale

-Figure 7 B: too small

Line 277: any suggestions about the other mechanisms of autophagy?

I7557 is more effective than MITC and I7447, please show results from other compounds  and othe rresearch groups that might ne similar.

Screening activity report (SAR) line 293

Line 315: remove or rephrase : As we have already done in our previous study.

section 4.6 : add more details about FACS parameters and channels.

-Avoid using We and replace by passive form.

-Discussion major point to be added: suggest mechanism causing effectiveness and differences between Imatinib and MITC{MITC derivatives, as well as signaling target possibilities and differences.

-A model to summarize the results of this study could be useful at the end of the conclusion section.

Author Response

Reviewer 3 Report (New Reviewer)

Two aspects must be addressed

1) In the viability test a reference compound is missing

2) Conclusions must be expanded about the significance of the results for future developments

Author Response

This manuscript is a resubmission of an earlier submission. The following is a list of the peer review reports and author responses from that submission.

Round 1

Reviewer 1 Report

The article of Lin J.-F. et al. contains introduction, results, discussion, materials methods, and conclusion sections. It is well organised and easy to understand.

 Minor revisions:

1. Would be better if authors could provide structural formula of 6-(methylsulfinyl)hexyl isothiocyanate and formulas of its new derivatives.

2. For the figures titles (above figs.) should be used standard style (A, B, C.., not 1a, 1b) with description below (For instance: Fig. 1. (A) - …, (B) - …)

3. Some figures could be put in supplementary material. Some data presented in the figures could be combined.

4. In introduction (lines 70-74)and discussion (lines 275-277) sections the authors write that chemical modifications of 6-MITC change of its antitumor activity, giving the reference on their previous work. Would be better if the authors in the present article describe more clearly which of the compounds is more active, why, and could other further chemical modifications cause increasing of its antitumor activity.

Author Response

Manuscript ID:  ijms-2255555

Dear editor:

It is our immense appreciation and honor to have the editor and reviewers putting the time and effort into reviewing the manuscript. The mentioned points have enabled us to improve and refine our work.

Based on the comments provided by reviewers, we revised the opinions from the reviewers. We re-submit this version for consideration of publication in Phytochemicals and Antioxidant, Anti-inflammatory and Cytotoxic Properties / Bioactives and Nutraceuticals /International Journal of Molecular Sciences.

Sincerely yours,

Yu-Jen Chen MD, PhD

Department of Radiation Oncology, MacKay Memorial Hospital,

No.92, Sec 2, Chung-Shan North Road, Chung-Shan Dist., Taipei 10449, Taiwan

Fax: (886) 2 2809 6180

Phone: (886) 2 2809 4661 ext. 2301

Reviewer Comments:

Reviewer 1:

The article of Lin J.-F. et al. contains introduction, results, discussion, materials methods, and conclusion sections. It is well organised and easy to understand.

 Minor revisions:

  1. Would be better if authors could provide structural formula of 6-(methylsulfinyl)hexyl isothiocyanate and formulas of its new derivatives.

Response: We appreciate the helpful comment. We have provided structural formula of 6-(methylsulfinyl)hexyl isothiocyanate and formulas of its new derivatives on the graphical abstract and figure 1.

Line 74 and 80:

Figure 1: Chemical structures of 6-MITC, I7447 and I7557

  1. For the figures titles (above figs.) should be used standard style (A, B, C.., not 1a, 1b) with description below (For instance: Fig. 1. (A) - …, (B) - …)

Response: We appreciate the helpful comment. In the manuscript, the figures titles have been used standard style.

  1. Some figures could be put in supplementary material. Some data presented in the figures could be combined.

Response: We appreciate the helpful comment. Some figures were put in supplementary material and some new figures were put in manuscript (figure 1 and figure 3).

  1. In introduction (lines 70-74)and discussion (lines 275-277) sections the authors write that chemical modifications of 6-MITC change of its antitumor activity, giving the reference on their previous work. Would be better if the authors in the present article describe more clearly which of the compounds is more active, why, and could other further chemical modifications cause increasing of its antitumor activity.

Response: We appreciate the helpful comment. We added description in the discussion.

Line 286-291: “According to the data of cell cycle and acridine orange stain, I7557 is more active than 6-MITC and I7447. There is no difference in the activity of apoptosis induction between I7447 and I7557. The possible cause of differential efficacy by these three compounds might be different oxygen atom numbers on the sulfide group. We might want to start focusing in the design and evaluation of a proper SAR to improve potency and elucidation of possible mechanism of action.”

Reviewer 2 Report

Peer review report

Ms. Ref. No.: ijms-2255555

Title: Anti-Cancer Effects of Oxygen-Atom-Modified Derivatives of Wasabi 2

Components on Human Leukemia Cells

Overall, I found the paper to be well written and organized. Chen and co-workers present biological evaluation of two 6-MITC, I7447 and I7557 as possible anticancer agents in CML K562 cell line. They study their cytotoxicity and effect on mitosis, autophagy and apoptosis. Evaluation is correct and according to literature procedure. Although the works seems interesting, the tested molecule are not potent, mechanism of action remains unknown and authors have already published recently really similar works in Biomedicine (Ref 13) and Molecules (ref 15) following the same similar approach with similar results in the same (K562 for 6-MITC) and different cancer cell line (oral SAS and OECM-1). Unfortunately, it is my opinion that publication of this work would lack of novelty at its current state due to previously published works, and results still need to be improved. Thus, I cannot recommend publication of this paper as its current state. I also came across with some questions that need to be addressed. I explain my concerns in more detail below. I would be happy to reconsider my decision if the authors can address all the necessary concerns.

1.     There is no information of the evaluated compounds. Authors specified they were purchased but some data regarding to their purity and mass or characterization should be provided and include in the manuscript. There is not even structures of any of the 3 molecules evaluated in the manuscript.

2.       Can the authors provide some inside about possible mechanism of action for these compounds? Is it touching any known biological pathway implicated in CML progression or acting any other way? The authors make a fairly complete introduction about BCR-ABL1, known driving factor of CML but, for instance, then there is not evaluation of this classic CML target, it protein expression, effect over it phosphorylation or over any other known or affected protein related with CML progression. I do believe that if the authors would like to keep moving forward with this kind of compounds, they might want to start focusing in the design and evaluation of a proper SAR to improve potency and elucidation of possible mechanism of action.

3.       It might be interesting for the authors to evaluate this compounds synergism with IM to improve potency and in IM resistant cell lines for instance.

Author Response

Manuscript ID:  ijms-2255555

Dear editor:

It is our immense appreciation and honor to have the editor and reviewers putting the time and effort into reviewing the manuscript. The mentioned points have enabled us to improve and refine our work.

Based on the comments provided by reviewers, we revised the opinions from the reviewers. We re-submit this version for consideration of publication in Phytochemicals and Antioxidant, Anti-inflammatory and Cytotoxic Properties / Bioactives and Nutraceuticals /International Journal of Molecular Sciences.

Sincerely yours,

Yu-Jen Chen MD, PhD

Department of Radiation Oncology, MacKay Memorial Hospital,

No.92, Sec 2, Chung-Shan North Road, Chung-Shan Dist., Taipei 10449, Taiwan

Fax: (886) 2 2809 6180

Phone: (886) 2 2809 4661 ext. 2301

Reviewer Comments:

Reviewer 2

Overall, I found the paper to be well written and organized. Chen and co-workers present biological evaluation of two 6-MITC, I7447 and I7557 as possible anticancer agents in CML K562 cell line. They study their cytotoxicity and effect on mitosis, autophagy and apoptosis. Evaluation is correct and according to literature procedure. Although the works seems interesting, the tested molecule are not potent, mechanism of action remains unknown and authors have already published recently really similar works in Biomedicine (Ref 13) and Molecules (ref 15) following the same similar approach with similar results in the same (K562 for 6-MITC) and different cancer cell line (oral SAS and OECM-1). Unfortunately, it is my opinion that publication of this work would lack of novelty at its current state due to previously published works, and results still need to be improved. Thus, I cannot recommend publication of this paper as its current state. I also came across with some questions that need to be addressed. I explain my concerns in more detail below. I would be happy to reconsider my decision if the authors can address all the necessary concerns.

  1. There is no information of the evaluated compounds. Authors specified they were purchased but some data regarding to their purity and mass or characterization should be provided and include in the manuscript. There is not even structures of any of the 3 molecules evaluated in the manuscript.

 Response: We appreciate the helpful comment.

Line 80: We have provided structural formula of 6-(methylsulfinyl)hexyl isothiocyanate and formulas of its new derivatives on the graphical abstract and figure 1.

 Figure 1: Chemical structures of 6-MITC, I7447 and I7557

  1. Can the authors provide some inside about possible mechanism of action for these compounds? Is it touching any known biological pathway implicated in CML progression or acting any other way? The authors make a fairly complete introduction about BCR-ABL1, known driving factor of CML but, for instance, then there is not evaluation of this classic CML target, it protein expression, effect over it phosphorylation or over any other known or affected protein related with CML progression. I do believe that if the authors would like to keep moving forward with this kind of compounds, they might want to start focusing in the design and evaluation of a proper SAR to improve potency and elucidation of possible mechanism of action.

Response: We appreciate the helpful comment. We put the data into the supplementary figure1.

Supplementary figure1:

The expression of cell cycle related proteins in K562 cells. Cells were treated by I7557. (0 – 15 mM) for 48 hours.

  1. p-Histone H3
  2. p-Chk-1
  3. p-Chk-2
  4. p-Cdc25c

  1. p-Cdc2
  2. Cyclin-B1
  3. p-Plk-1

Line 114-120:

For K562 cells treated with I7557, the further classification of G2/M arrest by using expression of phosphorylated Histone H3 showed increased expression, indicating arrest at mitosis phase (supplementary Figure 1A). I7557 treatment for 48 h down-regulated the expression of p-Chk1, p-Chk2, p-Cdc25c and p-Cdc-2 (supplementary Figure 1B-E) whereas the expression of p-plk1 was upregulated (supplementary Figure 1G). These results implicated the mechanistic insight of I7557 towards proteins related to mitosis phase arrest.

  1. It might be interesting for the authors to evaluate this compounds synergism with IM to improve potency and in IM resistant cell lines for instance.

Response: We appreciate the helpful comment.

Line 88-91 and 99-107:

Line 88-91: For the viability of IM-resistant K562 cells, IM had no inhibitory effect. The wasabi component 6-MITC and derivative I7557 suppressed the viability of IM-resistant K562 cells, indicating their effectiveness on both parent and IM-resistant K562 cells (Figure 3).

Line 99-107:

3A I7447                              3B 6-MITC

3C I7557

Figure 3: Effect of Wasabi compounds on viability of parent and Imatinib-resistant K562 cells. Cell viability was assessed using the trypan blue exclusion assay. K562 cells were treated with or without 0.2 mM Imatinib mesylate for 48 and 72 h. (A) I7447; (B) 6-MITC; and (C) I7557 cells. Data from three separate experiments are expressed as the mean ± standard error of the mean (SEM). N = 3 for each group. *P < 0.05, **P < 0.01, ***P < 0.001.
